# Innate Immune Cells and Toll-like Receptor–Dependent Responses at the Maternal–Fetal Interface

**DOI:** 10.3390/ijms20153654

**Published:** 2019-07-26

**Authors:** Andrea Olmos-Ortiz, Pilar Flores-Espinosa, Ismael Mancilla-Herrera, Rodrigo Vega-Sánchez, Lorenza Díaz, Verónica Zaga-Clavellina

**Affiliations:** 1Departamento de Inmunobioquímica, Instituto Nacional de Perinatología Isidro Espinosa de los Reyes, Ciudad de México 11000, Mexico; 2Departamento de Infectología e Inmunología, Instituto Nacional de Perinatología Isidro Espinosa de los Reyes, Ciudad de México 11000, Mexico; 3Departamento de Nutrición y Bioprogramación, Instituto Nacional de Perinatología Isidro Espinosa de los Reyes, Ciudad de México 11000, Mexico; 4Departamento de Biología de la Reproducción, Instituto Nacional de Ciencias Médicas y Nutrición Salvador Zubirán, Ciudad de México 14080, Mexico; 5Departamento de Fisiología y Desarrollo Celular, Instituto Nacional de Perinatología Isidro Espinosa de los Reyes, Ciudad de México 11000, Mexico

**Keywords:** placenta, chorioamniotic membranes, extra-embrionary tissues, maternal–fetal interface, human pregnancy, immunomodulation, Toll-like receptors

## Abstract

During pregnancy, the placenta, the mother and the fetus exploit several mechanisms in order to avoid fetal rejection and to maintain an immunotolerant environment throughout nine months. During this time, immune cells from the fetal and maternal compartments interact to provide an adequate defense in case of an infection and to promote a tolerogenic milieu for the fetus to develop peacefully. Trophoblasts and decidual cells, together with resident natural killer cells, dendritic cells, Hofbauer cells and other macrophages, among other cell types, contribute to the modulation of the uterine environment to sustain a successful pregnancy. In this review, the authors outlined some of the various roles that the innate immune system plays at the maternal–fetal interface. First, the cell populations that are recruited into gestational tissues and their immune mechanisms were examined. In the second part, the Toll–like receptor (TLR)–dependent immune responses at the maternal–fetal interface was summarized, in terms of their specific cytokine/chemokine/antimicrobial peptide expression profiles throughout pregnancy.

## 1. Innate Cell Populations at the Maternal–Fetal Interface

Pregnancy represents a temporal state in which the harmonic development of the fetus depends on a delicate balance between anatomic, endocrine, metabolic and immune factors. Particularly, immune tolerance to the fetal semi-allograft depends on specific changes within the mother’s reproductive tract aimed at avoiding fetal rejection by the maternal immune system, while maintaining a functional defensive response against pathogens [1]. Indeed, this tolerogenic state is well established since early pregnancy, especially within those maternal and fetal tissues that are in direct contact with each other, i.e., villous trophoblasts in contact with maternal blood immune cells, and chorioamniotic membranes overlying the maternal decidual layer.

Normal pregnancy is characterized by a mild controlled systemic inflammatory state [2]. While the adaptive immune response plays a more preponderant role during implantation, parturition or intrauterine infections, the cellular components of the innate immune response permanently patrol the maternal–fetal interface for the presence of foreign antigens that may jeopardize pregnancy. In fact, the placenta has been suggested as a pivotal component of the innate immune response based on its release of antimicrobial peptides and cytokines in response to distinct metabolic challenges such as infection, hypoxia and nutritional status, among others [3,4,5,6]. A second important site of innate immune responses during pregnancy is localized at the decidual layer. Throughout gestation, the decidua is an active site of chemokine synthesis attracting neutrophils, natural killer cells, dendritic cells, macrophages and T regulatory lymphocytes. During the first weeks of pregnancy, 70% of decidual leukocytes are natural killer cells, 20–25% are macrophages and approximately 2% are dendritic cells whereas, in contrast, neutrophils comprise approximately 70% of maternal peripheral blood leukocytes [7,8,9]. 

In this review, the importance of the involvement of innate immune cells in implantation, decidual–stromal interaction, angiogenesis and fetal growth was emphasized. Then, a section on Toll-like receptor (TLR)–mediated responses was included to show their pathogen– tissue and temporal–specific immune mechanisms during pregnancy. 

### 1.1. Natural Killer Cells

Natural killer (NK) cells are large lymphocytes with abundant cytolytic granules, in charge of cytokine production and the early detection of virally infected or transformed cells [7]. In humans, NK cells express CD16 (responsible for antibody–dependent cellular cytotoxicity), CD56 (important in cell–cell and cell–matrix interactions) and CD2 (involved in nanotube formation), while lacking typical T–cell antigen receptors such as CD3. Upon activation, NK cells express cytotoxic receptors including CD314, CD335, CD336 and CD337 [10]. 

Decidual NK (dNK) cells represent a subpopulation showing a CD56^bright^ or CD56^superbright^ and CD16^-^ phenotype, which can only be observed in 10% of peripheral NK (pNK) cells [11]. Moreover, dNK cells have a higher cytokine and angiogenic secretory profile and poor cytotoxic potential, despite having abundant intracellular granules and intact cytolytic machinery [12]. In fact, in murine and human models, the amount of perforin granules is similar between pNK and dNK cells, but their reactivity is diminished in the latter [12,13]. 

Low cytotoxicity in dNK cells can be explained by their particular receptor expression profile. They express the natural killer group 2 receptor (NKG2, which recognizes human leukocyte antigen (HLA)–E molecules), the immunoglobulin–like transcript 2 receptor (ILT-2, which recognizes HLA-G1) and the killer–cell immunoglobulin-like receptors (KIR, which identify polymorphic HLA-C) [14,15,16]. This receptor profile allows them to identify both villous and extravillous trophoblasts without generating a cytotoxic response, and they are therefore part of the strategies involved in maternal–fetal immune tolerance [17]. 

During the first trimester of pregnancy, pNK cells represent approximately 5–30% of circulating leukocytes, whereas dNK cells comprise 70% of decidual leukocytes [7,18]; the other 30% being composed of macrophages and T–cells [17]. After the second trimester, dNK cells become progressively less granulated and their numbers gradually decline until only a few, if any, remain in the term pregnant uterus [19,20]. 

Implantation is a critical stage that depends on endometrial decidualization and leukocyte migration to the endometrial stroma. NK cell invasion of this tissue is a critical part of the processes and is proposed to be comprised of three stages: (1) pNK cell proliferation, (2) their migration to the decidua and (3) their differentiation into dNK cells (Figure 1). 

The first stage is primarily driven by progesterone (P4) [21,22], which is key in NK cell survival during the menstrual cycle when, in the absence of fertilization, P4 levels descend as the *corpus luteum* involutes and dNK cells begin to die; NK cell apoptosis is detected by macrophages and leads to menstrual breakdown [17]. 

Although interleukin (IL)-2 is a common stimulating factor for NK cell proliferation, it is practically absent in first trimester decidua and placenta, therefore, is not considered to regulate dNK cell proliferation during pregnancy [23]. In contrast, IL-15 seems to be the most important regulatory cytokine for gestational NK cell proliferation because this cytokine is progesterone-dependent [22] and highly expressed in the human endometrium [24]. 

After pNK cell proliferation, migration into the decidual stroma takes place. This process depends on many trophoblastic, endometrial, endothelial, epithelial and stromal cell chemokines. These include Monocyte Chemotactic Protein 1 (MCP-1, CCL-2), Macrophage Inflammatory Protein 1-Beta (MIP-1β, CCL-4), Regulated Upon Activation, Normal T–cell Expressed and Secreted (RANTES, CCL-5), Monocyte Chemotactic Protein 3 (MCP-3, CCL-7), Macrophage Inflammatory Protein 3 Beta (MIP-3β, CCL-19), Secondary Lymphoid Tissue Chemokine (SLC, CCL-21), IL-8 (CXCL-8), Monokine Induced By Interferon-Gamma (MIG, CXCL-9) and the Interferon-Inducible Cytokine IP-10 (CXCL-10). Also, the human endometrium expresses fractalkine (CX3CL-1), the main NK cell chemoattractant [25,26,27]. Other hormonal factors such as estrogens, chorionic gonadotrophin (hCG) and prolactin also promote dNK cell migration to the mesometrial decidua and decidua *basalis* prior to implantation [7,21,28,29].

The third stage of NK cell invasion to the decidua is pNK cell differentiation into dNK cells. This depends on paracrine factors such as IL-11 or transforming growth factor beta (TGF-β), which are produced by the human endometrium, decidua and placenta, and favor conversion from the CD16^+^ to the CD16^-^ phenotype [30,31].

In summary, evidence suggests that during early pregnancy: (1) P4 and IL-15 contribute to pNK cell proliferation; (2) their migration to reproductive tissues is mediated by specific NK cell chemoattractants; and (3) the differentiation of pNK cells into dNK cells may be regulated by TGF-β and IL-11 through maternal/fetal paracrine pathways, which contribute to the lower cytotoxic phenotype observed in dNK cells.

An alternative hypothesis to explain dNK cell invasion of the decidua was proposed by Manaster and colleagues (2008). They suggested that an immature population of endometrial NK (eNK) cells that normally populate the non–pregnant endometrium become differentiated into dNK cells immediately after the hormonal stimulus of pregnancy [32]. Therefore, it seems that a progesterone–rich environment is the bridge for the two hypotheses, which remains to be elucidated.

Importantly, dNK cells participate in several vascular changes during pregnancy. Initial evidence by Guimond and colleagues (1997) showed that, in a murine model, a uterine environment depleted of dNK cells resulted in fetal death, in association with localized atherosclerosis and hypertension [33]. Other evidence indicates that dNK cells have a role in spiral artery remodeling [28,34]. 

The angiogenic role of dNK cells may be due to their elevated expression of various pro-angiogenic factors, including TGF-β, vascular endothelial growth factor (VEGF)-C, placental growth factor (PlGF) and angiopoietins 1 and 2 [35,36]. Indeed, dNK cell activation by IL-15 has been shown to increase VEGF-C expression, mainly during pregnancy [35]. Moreover, a paracrine pathway involving endometrium–produced estrogens and dNK cell–produced MCP-1 has been demonstrated in an in vitro angiogenesis assay. This implies an intercellular collaborative network that results in endometrial endothelium development [28].

### 1.2. Macrophages 

Macrophages are CD14^+^ and CD68^+^ phagocytes with a long life of months or even years. The main factor involved in their differentiation is the macrophage–colony stimulating factor (M-CSF), plus other mediators that vary depending on the specific type of macrophages being generated. Mature macrophages are 5 to 10 fold larger than monocytes, have more complex cytoplasmic organelles and acquire phagocytic activity [37]. They are involved in the removal of dead cells and debris during normal cellular cycles, as well as in the detection, ingestion or processing of foreign material during inflammation [38]. 

There are two main types of macrophages: M1 with microbicidal and inflammatory functions (classical activation); and M2 with immunomodulatory functions (alternatively activated macrophages) [37,39]. There are also other macrophage subpopulations with both M1 and M2 capacity, which secrete both pro– and anti–inflammatory cytokines, and therefore belong to an intermediate M1/M2 phenotype [40]. 

The non–pregnant endometrium has a low concentration of macrophages, with fluctuating numbers along the menstrual cycle [41]. After fertilization, macrophages represent the second most abundant decidual leukocyte population (around 20–30%). The decidual stroma is an active site for the synthesis of different chemokines which attract macrophages into the decidua, endometrium and myometrium. These chemokines include MCP-1, Macrophage Inflammatory Protein 1-alpha (MIP-1α, CCL-3), MIP-1β, MCP-3, Lymphocyte and Monocyte Chemoattractant (LCC-1, CCL-16), IP-10, Macrophage Inflammatory Protein 2 gamma (MIP-2γ, CXCL-14) and fractalkine [42]. 

An inflammatory M1 phenotype is associated with human implantation, characterized by the up–regulation of TNF-α and implantation–associated genes such as carbohydrate sulfotransferase 2 (CHST2), MIP-1β and Growth Regulated Oncogene alpha (GRO-α) [43]. Therefore, macrophages have been proposed as active molecular mediators between the embryo and endometrium in humans [44,45]. However, in primate and murine models, macrophages seem to oppose the implantation process [46,47]. These differences between species deserve to be further studied while considering the gestational window of action, macrophage phenotype and tissue–specific interactions. 

Following implantation, decidual macrophages increase markedly in the decidual stroma and are characterized by an M2 phenotype involving a high expression of IL-10 and indoleamine 2,3-dioxygenase (IDO), and a reduced expression of CD86, which is required for T–lymphocyte activation [48]. 

During the first 22 weeks of gestation, decidual macrophages control spiral artery remodeling, and after that time their numbers decrease in the decidual stroma [37,49]. Their presence has been characterized by immunohistochemistry in the immediate vicinity of spiral arteries with a disorganized endothelial layer in which extravillous trophoblasts are not yet present [37,50]. These macrophages secrete angiogenic factors such as VEGF, fibroblast growth factor (FGF), platelet derived growth factor (PDGF) and IL-10. Furthermore, macrophage production of nitric oxide and reactive oxygen species (ROS) has been identified as a mechanism for promoting endometrial angiogenesis in a bovine model [51]. 

Their pro–angiogenic factor production, together with the secretion of remodeling factors such as matrix metalloproteinases (MMP) -3 and -9 [43], suggests that decidual M1 macrophages modulate an inflammatory environment that results in increased trophoblastic invasion and spiral artery remodeling.

Decidual and endometrial macrophages also have a role in the removal by phagocytosis of apoptotic bodies from villous trophoblasts, which are normally released during villous tree remodeling. Phagocytosis helps to prevent the decidual tissue from releasing pro–inflammatory factors as a result of the accumulation of apoptotic bodies, which could affect pregnancy homeostasis [52]. 

Decidual macrophages have been suggested to participate in pregnancy maintenance by maintaining an immunomodulatory M2 phenotype throughout gestation. This phenotype changes at term when macrophages present a pro–inflammatory M1–type polarization and differentiate to dendritic cell–like cells with immunostimulatory capabilities [53,54]. At the end of pregnancy, macrophages also play a role in cervix remodeling during parturition [55], while in post–partum they clear the uterus of senescent cells and contribute to the recovery and maintenance of uterine tone and integrity [56]. 

#### Hofbauer Cells

Since the fourth week of pregnancy, a diverse group of fetal macrophages, designated as Hofbauer cells, begins to infiltrate the villous stroma of the placenta and reside in the mesenchymal stroma of villous trees adjacent to the fetal capillary network [57]. Based on their cellular structure, four different types of Hofbauer cells have been identified: Lamellipodia, funnel–like structures, bubbles or blebs, and microplicae [58].

Even though a great majority of maternal macrophages at the fetal–maternal interface derive from circulating monocytes, Hofbauer cells have different origins that vary depending on gestational age. During the first weeks of pregnancy, they may differentiate from villous mesenchymal stem cells or could be originated from hypoblast–derived cells [59,60]. From this point until parturition, Hofbauer cells may derive from fetal hematopoietic stem cells [59]. These cells do not express the proliferation marker protein Ki-67, so they do not experience mitotic division and therefore only increase in number through the previously mentioned differentiation pathways [61]. In a parallel way to macrophages of maternal origin, Hofbauer cells also present a peak concentration during the first trimester of pregnancy, which declines at term [39,61].

Hofbauer cells present an M2 phenotype with an anti–inflammatory cytokine expression profile (mainly IL-10 and TGF-β) [62], which helps to prevent fetal rejection and allows fetal growth until parturition [63]. Additionally, Hofbauer cells express M2 genes such as MCP-1, MCP-4 (CCL-13), CD209 and alpha-2-Macroglobulin (a2M), whereas M1 genes such as TLR-9, IL-1β or CD48 are silenced [58,64].

The M2 phenotype characteristic of decidual macrophages and Hofbauer cells is promoted by various mechanisms, such as the arginine pool reserve, the response mediated by HLA-G and the mesenchymal/stroma interaction. In the first mechanism, the arginine microenvironment that is critical for M1 differentiation is low during the first trimester of pregnancy, when placental tissue is rich in cytosolic arginase I and mitochondrial arginase II, both of which catalyze L-arginine to urea and L-ornithine [65,66]. Therefore, placental arginine levels are low enough to favor an M2 Hofbauer polarized phenotype [67]. 

The second mechanism involves the trophoblasts’ secretion of soluble HLA-G, which is detected by the ILT-2 receptor in macrophages. This contact promotes the differentiation of decidual macrophages to an M2 phenotype with increased expression of IDO and GRO-α [68,69]. 

In the third mechanism, the stromal microenvironment in villous trees also favors M2 polarization, although the process is not completely understood. Abumaree and colleagues (2013) observed that the co–culture of monocytes with placental mesenchymal stem cells resulted in a shift to M2–like macrophage differentiation, even though they were stimulated to follow the inflammatory M1 pathway [70]. Furthermore, Hofbauer cells isolated from their mesenchymal stromal matrix become M1–polarized in the absence of stroma [71], but do not change the M2 established phenotype when remaining in their enriched stromal matrix [72]. 

Maternal macrophages differ from Hofbauer cells in that the former confer protection against viruses and bacteria, while the latter are hyporesponsive to viruses. Indeed, cytomegalovirus, papillomavirus, herpes simplex and zika virus have all been detected within Hofbauer cells, and viral load correlates with neonatal morbidity [58,73,74,75,76]. Therefore, Hofbauer cells are not efficient at controlling viral infections.

On the other hand, one of the main functions of Hofbauer cells seem to be related to the paracrine control of placental angiogenesis and villous tree growth and branching. Hofbauer cells secrete high amounts of the pro-angiogenic molecules VEGF and FGF2, and culture media from these cells favor in vitro angiogenesis of fetal/placental endothelial cells [77]. Additionally, Hofbauer cells promote branching morphogenesis in chorionic villous trees by increasing the sprouty proteins (Spry) -1, -2 and -3, and develop a mesenchymal–epithelial interaction with trophoblasts [78]. It has also been suggested that Hofbauer cells participate in the regulation of stromal fluid balance, ion exchange and the transfer of serum proteins to the vascular system [39]. 

### 1.3. Dendritic Cells

Dendritic cells (DCs) have a critical role as a bridge between the innate and adaptive immune responses, and are present in the non-pregnant uterus, cycling endometrium, placenta and decidua. Less than 2% of CD45^+^ decidual leukocytes are DCs, and they are possibly the least well understood immune cell population during pregnancy.

Typically, DCs take up bacteria and antigens in the peripheral tissues, which induces a maturation program that includes loss of endocytosis activity and an up-regulated expression of co-activation molecules. After that, they migrate to the lymph nodes for presenting antigen-derived peptides to host T-cells in the context of a major histocompatibility complex (MHC) and drive a T helper (Th)-2, Th-1, Th-17 or regulatory T-cell (Treg) profile. 

Decidual DCs (dDCs), however, are a tolerant, immunorregulatory and heterogeneous population, phenotypically different from peripheral DCs. Decidual DCs are immature cells which express combinations of CD11c, HLA-G, CD14 and DC-SIGN (dendritic cell specific ICAM-grabbing non integrin, CD209). Nevertheless, all dDC subpopulations have in common a lower or null expression of T-cell activating markers CCR7, CD25, CD80, CD83, CD86, CD40 and CD205 [79].

From the early stages of implantation, migratory DCs populate the decidual stroma and modulate decidual angiogenesis and vascular expansion via the stromal cell derived factor 1 (SDF-1) and its receptor [80]. The angiogenic role of dDCs is most evident during implantation, when they are most abundant. After that time, their frequency declines progressively and remains low throughout pregnancy [81].

Some evidence has strengthened the idea that dDCs are an immature cell subpopulation during normal human pregnancy. Della Bella and colleagues (2011) showed that these cells exhibited a lower synthesis of IFN-γ, IL-10 and IL-12, lower levels of HLA-DR and an inhibitory activity upon T-cell proliferation [82]. Moreover, Bachy and colleagues demonstrated that monocytes isolated from pregnant women had a considerably lower differentiation rate to DCs than monocytes isolated from non-pregnant women. Once differentiated, these DCs conserved an immature phenotype in vitro [83]. 

On the other hand, observational studies have found a positive association between a higher frequency of mature peripheral or decidual DCs and the development of pregnancy-related inflammatory states such as preeclampsia and miscarriage [84,85]. This implies the need for a delicate balance between mature and immature dDCs for ensuring a healthy pregnancy.

The healthy pregnancy microenvironment seems to favor these immature characteristics of DCs in various ways, suggesting that its control is part of the mother–fetus–placenta tolerance mechanisms during pregnancy. Peripheral DCs treated with supernatants from decidual cell cultures or with serum from pregnant individuals show diminished T-cell activation capabilities [82,86,87]. In this regard, in vitro studies suggest that estradiol, soluble HLA-G, hCG and glycoprotein 1a are major pregnancy-derived candidates that limit DC maturation, stimulate type 2 T-cell responses and induce apoptosis of Th1 T-cells [86,88,89,90,91]. However, there are other placental- and decidual-derived secondary factors which can induce a tolerogenic profile in DCs, including prostaglandin E2, calcitriol (vitamin D_3_) and thrombopoietin [92,93,94]. Conversely, dehydroepiandrosterone sulphate (DHEAS) seems to favor DCs’ maturation and their ability to activate type 1 T-cell responses [95]. 

Moreover, recent evidence indicates that the placenta is also a potent paracrine inhibitor for decidual DCs’ maturation. Placenta-derived products may reduce the expression of CD40, CD80, CD83 and CD86 in DCs, limiting their ability to stimulate CD4^+^ T-cell proliferation even under an inflammatory stimulus [96,97]. These products may also promote a constitutive immature DC phenotype that is characterized by a higher expression ratio of IL-10 and TGF-β over IL-12 and TNF-α [98,99]. 

Additionally, placenta-derived products may favor a DC phenotype that induces an increased MHC class II expression, Th2 cytokine synthesis, and greater differentiation rates into CD4^+^CD25^+^FOXP3^+^ Treg in T-cells, while promoting reduced cytotoxicity in NK cells by inducing a CD56^bright^CD16^-^ dNK cell phenotype [100]. 

Regarding the migration of dDCs, there is an anatomical need for a physical separation between dDCs and lymphatic vessels. Indeed, the decidua creates a limitative environment for dDCs’ migration into lymphatic vessels, since the lymphatic vasculature is practically absent in human decidual stroma and is only present in the borderline of the non-decidualized endometrium. Interestingly, the collapse and active destruction of the lymphatic vasculature may be induced by the decidualization process itself [101]. 

Therefore, since dDCs are virtually trapped within the decidual stroma, evidence supports the paradigm that the detection of feto-placental alloantigens by T-cells is directed by the passive transport of antigens through lymphatic vessels into secondary uterine-draining lymph nodes. Fetal antigens are then taken up by local lymphatic DCs, which in turn present them to CD4^+^ T-cells to induce their proliferation [81]. These processes provide exciting new research areas involving the physiological functions of dDCs, including angiogenesis, decidualization, placentation, fetal growth and metabolism, which seem to be, at least in part, a collaborative work between dDCs and dNK cells [102]. 

Finally, DCs can selectively participate in the innate immune defense during pregnancy. Decidual DCs may not be effective at sensing bacteria and inducing T-cell responses, but they do express Toll-like receptors (TLR) -2 and -4 and are therefore capable of detecting Gram-positive and -negative bacteria, respectively [97]. Peripheral DCs, however, can recognize viral DNA through TLR-9 and produce type I interferons (IFN) such as IFN-α [103]. They can also actively fight against influenza A virus by inducing MHC-II, CD69, IP-10 and MIP-1β chemokines [104], and constitutively produce the microbicidal peptides alpha-defensins -1, -2 and -3 [105]. 

### 1.4. Neutrophils

In non-pregnant women, around 50–70% of peripheral blood leukocytes are neutrophils [106], while during pregnancy the percentage rises to 60–95% [9,107]. These elevated neutrophil numbers, a condition known as neutrophilia, are a well-characterized phenomenon during the second and third trimesters of pregnancy [9,107,108]. 

The fact that neutrophilia is a physiological condition during pregnancy can be understood by considering the various roles that neutrophils play in innate immune responses. Neutrophils are the first line of innate defense to protect both mother and fetus. They exert a variety of mechanisms to combat tissue damage and/or infections, including: (1) Phagocytosis; (2) production of granules with potent proteolytic activity; (3) synthesis of reactive oxygen species (ROS) and hypochlorous acid (HClO); (4) production of peptides with microbicidal action; (5) formation of neutrophil extracellular traps and secretion of pro-inflammatory cytokines. Moreover, neutrophils directly interact with DCs, macrophages, NK cells, T and B cells, therefore enhancing or down-modulating both the innate and adaptive immunities [109].

The possible causes that lead to neutrophilia during pregnancy have not been completely elucidated. Among the known factors that may stimulate neutrophil production are granulocyte-colony stimulating factor (G-CSF) and granulocyte/macrophage colony stimulating factor (GM-CSF), which are induced by estrogens or P4 in multiple cell types [110,111,112]. The estrogenic/progestagenic environment that prevails during pregnancy could explain the higher expression of these factors and consequent neutrophilia. 

An additional regulatory factor for neutrophil induction is the glycoprotein Human Neutrophil Antigen 2a (CD177), which is overexpressed in leukocytes during pregnancy [113]. In fact, alloimmunization against CD177 is a clinical cause of neonatal neutropenia [114,115]. 

Interestingly, G-CSF not only regulates neutrophil counts but also contributes to the stimulation of neutrophil extracellular trap (NET) formation during pregnancy [116]. NETs are interlaced networks composed of a chromatin backbone, histones and granular proteins superimposed over an extracellular matrix. To form NETs, neutrophils rupture their nuclear envelope and plasma membrane to release decondensed chromatin and granule contents. This forms a mass with bactericidal properties that allows the catching of fungi or bacteria, which are later destroyed via histones and toxic granules or by other innate immune cells, including NK cells and macrophages [117]. 

A form of cell death mediated by NETs, called NETosis, has been characterized during pregnancy as being upregulated by hCG and estradiol, while down-regulated by P4 [116]. The microbicidal activity of NETs has been observed in amniotic fluid from women with clinical chorioamnionitis and is therefore considered an additional defense mechanism during intrauterine infections [118].

As previously mentioned, the success of embryo implantation depends on the interplay of various immune cell types, and this includes neutrophils [119,120,121]. During implantation, the P4 stimulus induces IL-8 and MCP-1 chemokines in endometrial epithelial cells [122], thereby favoring neutrophil and monocyte recruitment. Moreover, this IL-8-dependent neutrophil recruitment seems to be controlled in a paracrine way since it is influenced by trophoblastic contact [123].

While in non-pregnant women neutrophils are regularly recruited to the cervix, endometrium and Fallopian tubes [8,124,125], during pregnancy these cells also invade the decidua, chorioamniotic membranes and placenta [111,126,127]. Neutrophil roles in decidua are related to angiogenesis, spiral artery remodeling and extravillous trophoblast migration [128,129]. 

As with other innate immune cells, the decidual microenvironment also confers phenotypic differences between decidual and peripheral neutrophils. Immunohistochemical and cytometric analyses revealed that decidual neutrophils can infiltrate into the decidua *basalis* (i.e., the placenta), presenting a higher expression of CD66b, VEGF-A, arginase-1 and MCP-1 which confer neutrophils a pro-angiogenic profile with enhanced diapedesis and favors their endothelial extravasation for uterine vascular remodeling [130]. 

Decidual neutrophils also have a pivotal role during implantation and trophoblast migration, which is related to their pro-degradative activity. Indeed, neutrophils synthesize MMP-9 and neutrophil gelatinase-associated lipocalin (NGAL) [121]. The latter is increased during pregnancy [131]. Both proteins form a complex where NGAL protects MMP-9 from degradation, preserving its enzymatic activity and thus aiding in extracellular matrix remodeling and trophoblast invasion [132].

Finally, some reproductive diseases such as preeclampsia, infertility or fetal loss have been associated with excessive neutrophil activation, inadequate NET formation or neutrophil migration [126,133,134]. Due to this, G-CSF treatment has been used in clinical practice in order to achieve better results in assisted fertility rates and outcomes in women with a thin endometrium or recurrent implantation failure [135,136,137]. However, both great caution and the patient’s personal characteristics must be taken into account when considering these treatments due to the various known unwanted side effects of G-CSF. These include mucositis, splenic enlargement, hepatomegaly, transient hypotension, epistaxis, urinary abnormalities, osteoporosis, exacerbation of rheumatoid arthritis, anaemia and pseudogout [138].

## 2. TLR-Dependent Immune Responses During Pregnancy

### 2.1. Toll-Like Receptors: Structures and Signaling Pathways

Toll-like receptors (TLRs) are type I transmembrane proteins with a leucine-rich extracellular domain, which sense three types of ligands: Nucleic acids, proteins and lipids. In humans, ten TLRs (TLR1 to TLR10) and one pseudogene (TLR11) have been identified, which may be classified depending on their cellular localization into cell surface TLRs and endosomal TLRs. In turn, such localization is related to their sensing capacity. As TLR3, TLR7, TLR8 and TLR9 are involved in viral recognition, they are located in endosomes, instead of the cellular membrane [139].

TLRs belong to the pattern recognition receptor (PRR) family, which are in charge of the innate recognition of pathogen-associated molecular patterns (PAMPs). PAMPs are conserved sequences of proteins, lipids, polysaccharides, DNA or RNA present in the membrane or envelope of pathogenic microorganisms. TLRs are also capable of recognizing damage-associated molecular patterns (DAMPs), such as heat shock proteins among other proteins that are released by cells undergoing stress-dependent apoptosis [140]. 

The main agonists identified for each TLR are summarized in the first columns of Table 1. Most TLRs make homodimers to sense ligands. Only TLR1 and TLR6 make heterodimers with TLR2 to sense ligands. TLR1/TLR2 heterodimers recognize triacylated lipopeptides whereas TLR6/TLR2 heterodimers recognize diacylated lipopeptides [141]. 

After ligand/receptor binding, TLRs are capable of initiating two different signaling pathways, both of which end in a pro-inflammatory cascade. One is dependent and the other independent of the myeloid differentiation factor 88 (MyD88) (Figure 2).

All TLRs, except TLR3, can activate the MyD88-dependent pathway. Classically, TLRs’ homo- or hetero-dimers can directly recognize PAMPs, as has been previously described. However, TLR4 and MD-2 form a dimer without Lipid A, then Lipid A (core of LPS) binds to MD-2 complexed with TLR4 in CD14-dependent manner [142,143]. 

TLR1, TLR2, TLR4 and TLR6 require the TIR domain containing Adaptor Protein (TIRAP) to activate the MyD88-dependent pathway. Canonical MyD88-dependent signaling involves a phosphorylation sequence starting with IL-1 receptor-associated kinases (IRAK)-4 and -1/2, followed by TNF receptor-associated factor (TRAF6) and finally the IκB Kinase (IKK) complex, conformed by IKKα, IKKβ and the NF-κB Essential Modulator (NEMO). IKK phosphorylates IκB proteins. This results in IκB ubiquitination and degradation. Consequently, IκB is dissociated from NF-κB, which can then translocate freely to the nucleus and modulate the expression of genes containing NF-κB binding sites, including the main pro-inflammatory cytokines TNF-α, IL-6 and IL-8 [144]. This pro-inflammatory pathway increases the synthesis of antimicrobial peptides, cytokines and chemokines to restrain and fight against pathogen infection. 

The MyD88-independent pathway is activated only by TLR3 and TLR4. MyD88-independent signaling by TLR4 requires the Translocation Associated Membrane Protein (TRAM) for its association with the TIR Domain-Containing Adapter Protein Inducing IFN-β (TRIF). TLR3 can also activate TRIF, and then it phosphorylates the complex formed by IKKε, IKKι and TANK binding kinase 1 (TBK1). This activates Interferon Regulatory Factor 3 (IRF3), which then translocates to the nucleus and activates the transcription of interferons alpha and beta, as well as other interferon-induced genes [145]. Additionally, TLR7 and TLR9 induce IFNα/β production via the MyD88/IRF7 pathway. 

A special case is that of TLR10, which is still called an orphan receptor since it remains without a well-characterized ligand or function. However, genomic studies suggest that TLR10 is related to TLR1 and TLR6 [139], and recent evidence suggests that it may recognize double-stranded RNA [146]. Interestingly, TLR10 does not activate the immune system and more likely acts as an anti-inflammatory pattern-recognition receptor that may suppress inflammatory signals [147]. 

As this review has shown, each one of the known TLRs recognizes specific PAMPs and triggers a particular cytokine, chemokine and/or antimicrobial peptide response. This necessarily changes the concept of the innate immune response as one of general or unspecific nature and gives rise to the idea of a specific signaling network associated with particular pathogens and particular cell types that are involved in such response. 

### 2.2. TLRs During Pregnancy

Throughout pregnancy, all TLRs are expressed at the maternal–fetal interface in a spatio-temporal specific manner [148]. TLR expression patterns in human trophoblasts are summarized in Table 1. 

First trimester trophoblasts do not express TLR6 while it is present in term trophoblasts. However, the opposite is observed in relation to TLR1, which is present in first trimester and absent in term placentas [149,150]. This opposite pattern of expression between TRL6 and TLR1 could be related to their sensibility for differentiating triacylated from diacylated lipopeptides when forming the heterodimers TLR2/TLR1 or TLR2/TLR6. 

TLR2 and TLR4 are expressed in syncytiotrophoblasts throughout pregnancy, with the highest expression during the third trimester [151,152], possibly to restrain cervico-vaginal Gram-positive and -negative infections, which could compromise pregnancy continuity in this critical period. Viral sensors TLR3, TLR7, TLR8 and TLR9 are mainly expressed at the syncytial layer and the amniotic epithelium [153].

It is noted that a very high and polarized TLR2 protein expression has been found at the outer (apical) plasma membrane adjacent to the syncytiotrophoblast layer at 6–7 weeks of gestation. This TLR2 abundance, together with TLR3 and TLR4, is so high that it resembles a tight TLR barrier forming a defensive wall along the cytotrophoblast layer [154]. Interestingly, this TLR expression pattern is only seen during the first trimester and becomes attenuated afterwards, suggesting that the placenta intensely protects the fetus from pathogen attacks during early developmental phases, coinciding with the most vulnerable time of pregnancy [154]. However, such high TLR expression in the first trimester may also imply that an exacerbated inflammatory immune response could take place in the event of infection, therefore endangering the pregnancy in itself.

The underlying mechanisms that control spatio-temporal TLR regulation at the maternal–fetal interface still need to be elucidated. Special caution must be taken when using established placental cell lines to evaluate TLRs’ responses in placenta, since their TLR expression patterns differ from that observed in primary trophoblast cells. This is especially true for TLR2, which is practically absent in BeWo, JAR, JEG-3, AC1M-32, ACH-3P and HTR-8/SVneo cell lines [156].

During pregnancy, not only trophoblasts express TLRs as an innate defense tool, but also Hofbauer and endothelial cells, decidua and chorioamniotic membranes. Furthermore, a soluble form of TLR2 is also present in amniotic fluid [157], which diminishes pro-inflammatory signaling by ligand sequestering and consequently reduces TLR2 activation in the amniotic epithelia. 

As mentioned earlier, Hofbauer cells are generally considered to be inefficient at controlling viral infections, even though they express TLR3 which recognizes double-stranded viral RNA [57]. Nevertheless, an enhanced secretion of IL-6 and IL-8 by Hofbauer cells has been observed after treatment with Poly (I:C), a synthetic analog of double-stranded RNA (dsRNA). This treatment revealed that even though these are M2-type macrophages, their inflammatory responses were induced through TLR3, suggesting an important role of these cells in controlling viral infections [57].

On the other hand, Hofbauer cells actively express TLR2 and TLR4 [57,158], which allows them to readily respond against Gram-positive and Gram-negative bacteria, respectively. In fact, Hofbauer cells from women with chorioamnionitis overexpress TLR4, possibly to amplify the inflammatory signals that restrain bacterial infections [158]. At present, there are no available data regarding the expression of TLR1, TLR5, TLR6, TLR7, TLR8, TLR9 or TLR10 in Hofbauer cells. 

In decidual stroma from the first trimester and term, both mRNA and protein expressions have been identified for TLR1 to TLR10 [159,160]. Interestingly, the first trimester decidua produced twice or more the amount of mRNAs for most TLRs in comparison with the term decidua, with the exception of TLR2, TLR7 and TLR9 whose expression remained constant throughout pregnancy [160]. This enriched TLR pattern is probably related to the need for a safer environment in the highly active early decidua, whereas during the third trimester the placenta takes over the role for monitoring and controlling microbial antigen exposure. Clinically, TLR2 up-regulation has been demonstrated in decidual tissue from infection-associated preterm deliveries [161].

Regarding the chorioamniotic membranes, they express all ten TLRs throughout pregnancy [162,163]. In particular, TLR4 expression on the apical side of the amniotic epithelium enables it to react to pathogen presence in the amniotic fluid. This TLR4 expression, together with TLR2, is induced by chorioamnionitis [164]. Furthermore, the authors have observed in our laboratory that P4 diminished chorioamniotic expression of both TLR4 and MyD88. This is possibly a compensatory mechanism to limit an overproduction of pro-inflammatory cytokines in the event of an inflammatory stimulus, which might put the mother or fetus in danger [165].

The physiological relevance of TLR presence in gestational tissues resides in the fact that an eventual infection in the genitourinary tract could interrupt pregnancy by an untimely development of labor-related signals, causing premature rupture of membranes, which is strongly associated with intrauterine bacterial infections [166]. 

Interestingly, not only is there a precise spatio-temporal expression pattern of TLRs at the maternal–fetal interface, but there is also a specific pattern of inflammatory mediators and effector molecules triggered by particular PAMPs [162]. In Table 2, the cytokine, chemokine and antimicrobial peptide cascades resulting from PAMPs’ interaction with specific TLRs in a tissue-dependent manner was summarized. 

The specificity of the TLR-dependent innate immune responses can be observed in various cases involving the in vitro stimulation of the chorionic layer of fetal membranes. TLR4 activation by *Gardnerella vaginalis* resulted in higher IL-10 secretion but no change in TNF-α, whereas LPS/TLR4 activation in the same tissue resulted in TNF-α induction but minimal change in IL-10 [167,168,169,170]. Similarly, TLR2 activation by *Candida albicans* induced a higher production of IL-10 in comparison with that observed when TLR2 was activated by *Streptococcus agalactiae* or lipoteichoic acid from *Streptococcus pyogenes* [167,171]. Therefore, it seems that IL-10 anti-inflammatory synthesis is differentially magnified in response to a particular yeast.

Nowadays, additional TLR roles have been observed that are unrelated to immune recognition. For example, TLR7 activation negatively regulates neuronal differentiation and dendrite growth through the MyD88/c-Fos/IL-6 canonical pathway. Consequently, it has been proposed that viral activation of TLR7 could be an additional factor implicated in deleterious neurological development and neonatal cognition in children exposed to a viral infection during pregnancy [179].

Also, there are recent reports of TLR-dependent angiogenesis regulation. In first trimester cultured trophoblasts, synthesis of the pro-angiogenic factor PlGF was induced by TLR2 agonists, while inhibited by the LPS/TLR4 pathway. On the other hand, ligands for virus-sensing TLRs, namely TLR8 and TLR9, stimulated the secretion of the anti-angiogenic molecule sFlt-1. Hence, deleterious angiogenic signaling could be modulated during viral or bacterial infection in pregnancy and therefore deserves special clinical attention [180].

## 3. Summary

Local and systemic immune blood cells contribute to the establishment of an immunological environment at the maternal–fetal interface that helps maintain a healthy pregnancy. Specifically, decidual NK cells and macrophages, together with trophoblasts, not only promote oxygen and nutrient delivery by acting on uterine spiral arteries, but also facilitate angiogenesis.

Neutrophils are also pro-angiogenic and, together with macrophages, are sources of pro-inflammatory cytokines at the maternal–fetal interface during critical stages of pregnancy. These innate immune cells can also phagocytize bacteria and apoptotic bodies, while exerting microbicidal activity either by NET formation, modulation of T-cell activity or secretion of cytokines.

Hofbauer cells and decidual macrophages also display phagocytosis and elicit anti-inflammatory activity. They maintain an M2 phenotype throughout pregnancy, which changes to the pro-inflammatory M1-type at term.

Antigen-presenting macrophages and DCs are important against infection and modulate trophoblast invasion, angiogenesis and vascular remodeling. A summarized scheme of the angiogenic role of innate immune cells during pregnancy is presented in Figure 3. Thus, all these cells conjointly are key actors in the immune milieu during implantation, fetal development, parturition and upon infection.

Finally, all TLRs are expressed at the maternal–fetal interface in a spatio-temporally specific manner. There is also a specific pattern of inflammatory mediators and effector molecules triggered by particular PAMPs which results in specific cytokine, chemokine and antimicrobial peptide cascades in a tissue-dependent manner.

## Figures and Tables

**Figure 1 ijms-20-03654-f001:**
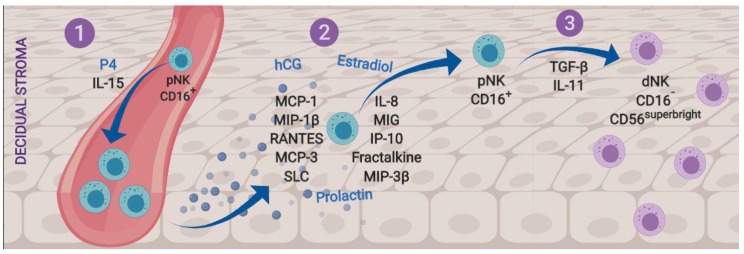
Stages of recruitment and differentiation of decidual natural killer (NK) cells. (1) Progesterone (P4) and interleukin (IL)-15 induce pNK cell proliferation. (2) Chemotactic factors produced by trophoblasts, endothelium and decidual epithelium selectively recruit pNK cells into the implantation site. (3) The endometrial environment, characterized by high levels of transforming growth factor beta (TGF-β) and IL-11, induces the differentiation of pNK cells to dNK cells, with the less cytotoxic phenotype CD16^-^ and CD56^superbright^. dNK = decidual natural killer cell; pNK = peripheral natural killer cell.

**Figure 2 ijms-20-03654-f002:**
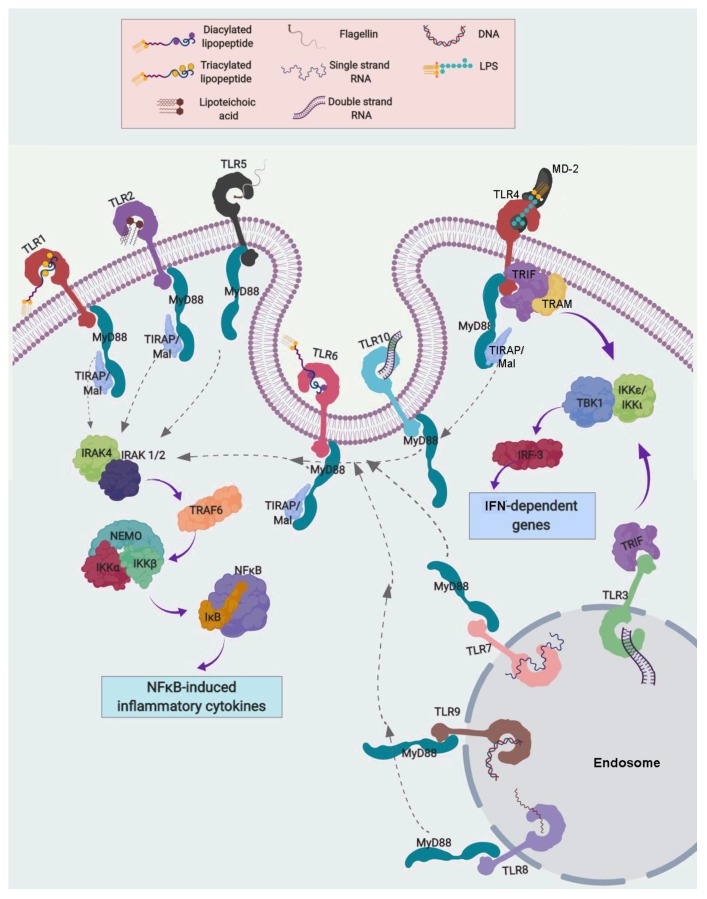
Toll-like receptor (TLR) signaling. Myeloid differentiation factor 88 (MyD88)-dependent and -independent pathways. The main ligands associated to TLRs are indicated, and required their TLR homo- or hetero-dimerization for its activation. All TLRs, except TLR3, can activate the MyD88-dependent pathway. TIR domain containing Adaptor Protein (TIRAP) is associated to TLR1, TLR2, TLR4 and TLR6 and can activate the MyD88-dependent pathway, which involves the activation in cascade of IL-1 receptor-associated kinase (IRAK)-4, IRAK-1/2, TNF receptor-associated factor (TRAF)-6, the IκB Kinase (IKK) complex and IκB proteins, which results in IκB ubiquitination and degradation. Then, NF-κB can freely translocate to the nucleus and modulate the expression of genes containing NF-κB binding sites. The MyD88-independent pathway is activated only by TLR3 and TLR4 through their association with the TIR Domain-Containing Adapter Protein Inducing IFN-β (TRIF) and the Translocation Associated Membrane Protein (TRAM), which phosphorylates the IKKε-IKKι-TBK1 complex. This finally activates interferon regulatory factor 3 (IRF3), which activates IFN-α and -β transcription, as well as other interferon-induced genes.

**Figure 3 ijms-20-03654-f003:**
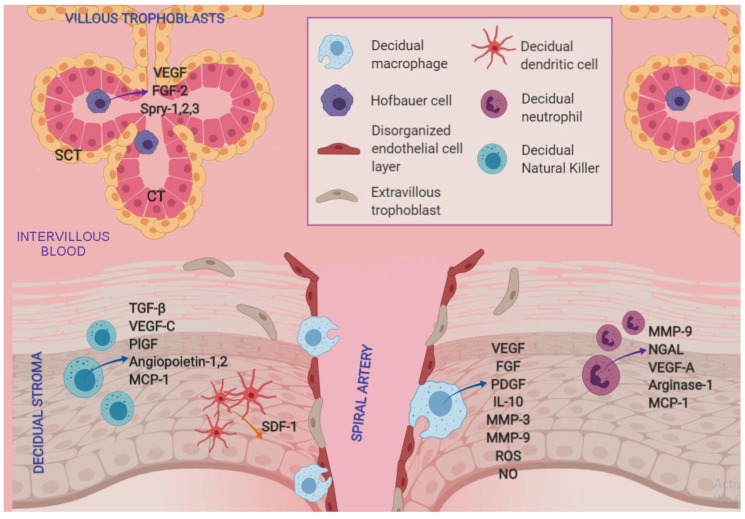
Pro-angiogenic network mediated by innate immune cells at the maternal–fetal interface. Angiogenic molecules secreted by macrophages, dendritic cells, natural killers and neutrophils in the decidua contribute to spiral artery remodeling and the replacement of endothelial cells with extravillous trophoblasts during trophoblast migration. Hofbauer cells immersed in villous mesenchyme produce VEGF, FGF-2, Spry-1, Spy-2 and Spry-3 to regulate fetal capillary growth, proliferation and branching. SCT = syncytiotrophoblasts; CT = citotrophoblasts.

**Table 1 ijms-20-03654-t001:** TLRs’ temporal patterns of expression in human trophoblasts.

	Main Ligands	TLR Expression			References
		1st Trimester	2nd Trimester	3rd Trimester	
**TLR1**	Triacylated lipopeptides and lipoproteins from Gram-positive bacteria. When forming a heterodimer with TLR2, it recognizes peptidoglycan and triacylated lipoproteins	++	++	-	[149,150,154]
**TLR2**	Lipoteichoic acid and lipoproteins from Gram-positive bacteria, lipoarabinomannans from mycobacteria and zymosan from yeast. Diacylated or triacylated lipopeptides depending on heterodimerization patterns	++	++	+++	[150,152]
**TLR3**	Double strand RNA	++	++	+++	[153,154]
**TLR4**	Lipopolysaccharides from Gram-negative bacteria, paclitaxel, heat shock proteins, heparan sulphate, reactive oxygen species (ROS), fibrinogen and fibronectin	++	+++	+++	[150,151,152]
**TLR5**	Flagellin from both Gram-positive and Gram-negative bacteria	++	++	++	[154,155]
**TLR6**	Diacylated lipopeptides from mycoplasmas (also heterodimerizes with TLR2)	-/+	+	++	[149,150,154]
**TLR7**	Single strand RNA and small synthetic compounds such as guanosine analogs or imidazoquinoline	+	++	+	[153,154]
**TLR8**	Single strand RNA and small synthetic compounds such as guanosine analogs or imidazoquinoline	+	++	+++	[153,154]
**TLR9**	Viral single-strand unmethylated CpG DNA and also fetal DNA	-	+	+	[153,154,155]
**TLR10**	Double strand RNA	+	+	+	[154]

Expression is presented in arbitrary units: - undetectable, + low expression, ++ moderate expression, +++ high expression.

**Table 2 ijms-20-03654-t002:** Cytokines, chemokines and antimicrobial peptides modulated by different TLR agonists in gestational tissues.

**Cultured Term Placenta**														
	**Cytokines**					**Chemokines**					**Antimicrobial Peptides**			
**Agonist of:**	**IL-6**	**IL-1β**	**TNF-α**	**IL-10**	**IL-2**	**IL-8**	**MCP-1**	**GRO-α**	**RANTES**	**MIP-1α**	**HBD1**	**HBD2**	**HBD3**	**Reference**
**TLR2**														
Lipoteichoic acid	++	++		+		++	+			+++				[172]
**TLR3**														
High MW Poly I:C	+					+								[153]
Low MW Poly I:C	++					++	+	+	+					[153]
**TLR4**														
LPS from *E. coli*						+	+	+						[173]
LPS from *E. coli*	+	+++	++	+		++				++				[174]
LPS from *E. coli*											UC	++	UC	[6]
LPS from *E. coli*	++	+++		+++		++	+			+++				[172]
**TLR7**														
Imiquimod	UC					+								[153]
**TLR8**														
ssRNA40	++					++								[153]
**TLR9**														
ODN21798	UC					UC								[153]
**Cultured Term Chorion**														
	**Cytokines**					**Chemokines**					**Antimicrobial Peptides**			
**Agonist of:**	**IL-6**	**IL-1β**	**TNF-α**	**IL-10**	**IL-2**	**IL-8**	**MCP-1**	**GRO-α**	**RANTES**	**MIP-1α**	**HBD1**	**HBD2**	**HBD3**	**Reference**
**TLR2**														
*Streptococcus agalactiae*				+										[167]
*Streptococcus agalactiae*											UC	+	UC	[175]
Lipoteichoic acid from *Streptococcus pyogenes*				+										[171]
*Candida albicans*				+++										[167]
**TLR3**														
High MW Poly I:C	+					+								[153]
Low MW Poly I:C	++					++								[153]
**TLR4**														
LPS from *E. coli*	UC	++	+++	+										[169]
LPS from *E. coli*	++	+++	+++	+		+								[170]
LPS from *E. coli*				++										[171]
*Escherichia coli*											UC	+	UC	[176]
*Gardnerella vaginalis*				+++										[167]
*Gardnerella vaginalis*	++	++	UC								UC	+	UC	[168]
**TLR7**														
Imiquimod	UC					+								[153]
**TLR8**														
ssRNA40	+					++								[153]
**TLR9**														
ODN21798	UC					UC								[153]
**Cultured Term Amnion**														
	**Cytokines**					**Chemokines**					**Antimicrobial Peptides**			
**Agonist of:**	**IL-6**	**IL-1β**	**TNF-α**	**IL-10**	**IL-2**	**IL-8**	**MCP-1**	**GRO-α**	**RANTES**	**MIP-1α**	**HBD1**	**HBD2**	**HBD3**	**Reference**
**TLR2**														
Peptidoglycans from *Staphylococcus aureus*	++	++		++	UC	++	UC	UC	+++	+				[162]
**TLR4**														
LPS from *E. coli*	UC	+++	+++	+++	UC	+	UC	++	+++	UC				[162]
**TLR5**														
Flagellin from *Salmonella typhimurium*	+++	+++	+++	UC	UC	+++	UC	+	+++	+				[162]
**TLR9**														
ODN21798	UC	UC	UC	UC	UC	UC	+	UC	-	-				[162]
**Cultured Term Chorioamniotic Membranes**														
	**Cytokines**					**Chemokines**					**Antimicrobial Peptides**			
**Agonist of:**	**IL-6**	**IL-1β**	**TNF-α**	**IL-10**	**IL-2**	**IL-8**	**MCP-1**	**GRO-α**	**RANTES**	**MIP-1α**	**HBD1**	**HBD2**	**HBD3**	**Reference**
**TLR2**														
Peptidoglycans from *Staphylococcus aureus*	++	++		++	UC	++	UC	UC	+++	+++				[162]
**TLR4**														
LPS from *E. coli*	UC	+++	+++	+++	UC	+	UC	++	+++	UC				[162]
**TLR5**														
Flagellin	+++	+++	+++	UC	UC	+++	UC	+	+++	+++				[162]
**TLR9**														
ODN21798	UC	UC	UC	UC	UC	UC	+	UC	-	-				[162]
**Cultured Term Decidual Cells**														
	**Cytokines**					**Chemokines**					**Antimicrobial Peptides**			
**Agonist of:**	**IL-6**	**IL-1β**	**TNF-α**	**IL-10**	**IL-2**	**IL-8**	**MCP-1**	**GRO-α**	**RANTES**	**MIP-1α**	**HBD1**	**HBD2**	**HBD3**	**Reference**
**TLR2**														
Lipoteichoic acid from *Streptococcus pyogenes*						++				++				[177]
Lipoteichoic acid from *Streptococcus pyogenes*				+++										[171]
*Group B Streptococci*						++				++				[177]
**TLR4**														
LPS from *E. coli*						+++				+++				[177]
LPS from *E. coli*				+++										[171]
**Cultured Term Placental Blood Mononuclear Cells**														
	**Cytokines**					**Chemokines**					**Antimicrobial Peptides**			
**Agonist of:**	**IL-6**	**IL-1β**	**TNF-α**	**IL-10**	**IL-2**	**IL-8**	**MCP-1**	**GRO-α**	**RANTES**	**MIP-1α**	**HBD1**	**HBD2**	**HBD3**	**Reference**
**TLR4**														
LPS from *E. coli*		++	++	UC		++				++				[178]
**Cultured Term Hofbauer Cells**														
	**Cytokines**					**Chemokines**					**Antimicrobial Peptides**			
**Agonist of:**	**IL-6**	**IL-1β**	**TNF-α**	**IL-10**	**IL-2**	**IL-8**	**MCP-1**	**GRO-α**	**RANTES**	**MIP-1α**	**HBD1**	**HBD2**	**HBD3**	**Reference**
**TLR2**														
Peptidoglycans from *Staphylococcus aureus*	UC					UC								[57]
**TLR3**														
Poly I:C	++					+								[57]
**TLR4**														
LPS from *E. coli*	+++					+								[57]

Compared to control: UC = unchanged, – = inhibition, + = 1–5 fold induction, ++ = 5–10 fold induction, +++ = over 10 fold induction. LPS = Lipopolysaccharide; MW = molecular weight; ss = single-strand.

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
