# Peer review of "Innate Immune Cells and Toll-like Receptor–Dependent Responses at the Maternal–Fetal Interface"

_ijms, 2019, doi:10.3390/ijms20153654_

Round 1

Reviewer 1 Report

Good and current review of innate immune defence processes operating at the materno-fetal interface including highlighting where we don't know anything but it would be good to know. There are quite a few English language usage errors in the manuscript that I have captured in my comments below. My other queries and comments are interspersed amongst these. The manuscript reads as though different parts are written by different authors. One author should harmonise this taking into account all of the corrections and queries above.

Line 14 – fetus exploit several mechanisms

Line 15- the nine

Line 19 – Hofbauer cells and other macrophages [neutrophils should be omitted and can be covered in ‘among others’]

Line 28 – the fetus

Line 30 – the fetal

Line 32 – pathogens.

Line 32 – the early

Line 33/34 – remove ‘between them’

Line 34 – embedded should be replace by bathed

Line 35 – the maternal

Line 37 – should adaptive be adaptive immune response?

Line 38 – deletion should be dilation

Line 39 – innate immune response

Line 40 – gravidity is the wrong word – it means the number of times a woman has been pregnant

Line 40 - The fetomaternal

Line 41 – innate immune activity

Line 41 – the placenta

Line 42 – cells of fetal

Line 44 – nutrimental should be nutritional 

Line 45 – innate immune responses

Line 45 – the decidua

Line 46 – the decidua

Line 47 – attract should be attracts

Line 47 – regulator should be regulatory

Line 50 – circulant should be circulating

Line 50 – innate immune cell

Line 52 – temporality should be temporally

Line 52 – ‘aborded to shown’ should be ‘included to show’

Line 55 – NK cells

Line 56 – virally infected or transformed

Line 56 – In humans,

Line 59 – remove ‘T-marker’

Line 60 – NK cells

Line 61 – Unlike peripheral NK (pNK) cells, decidual NK (dNK) cells represent a NK cell

Line 62 – dNK cells

Line 63 – pNK cells

Line 66 – dNK cells

Line 66 – pNK cells

Line 67 – cytotoxicity

Line 67 – dNK cells

Line 73 – pNK cells

Line 74 – dNK cells

Line 75 – macrophages and T cells

Line 75 – dNK cells

Line 75 – lesser should be less

Line 76 – with only a few remaining, if any, in the 

Line 79 – NK cell invasion

Line 80 – pNK cell; dNK cells

Line 81 – remove ‘during pregnancy’

Line 81 – The role of progesterone in NK cell surveillance

Line 82 – the menstrual

Line 82 – fecundation should be fertilisation

Line 83 – descend as the; dNK cells

Line 84 – lead to 

Line 85 – NK cells

Line 86 – and placenta

Line 87 – dNK cells

Line 88 – NK cell

Line 90 – pNK cell

Line 91 – remove ‘the’ at end of this line.

Line 92 – all chemokines should have their chemokine nomenclature included, e.g. IL-8 should have CXL8 mentioned as well

Line 93 – RANTES = Regulated upon Activation, Normal T cell Expressed, and Secreted

Line 96 – (IP-10)

Line 98 – dNK cells

Line 99 – what is meant by migration of dNK cells to smooth muscle cells?

Line 100 – Thirdly, pNK cell

Line 100 – dNK cells

Line 102 – extra space before Overall

Line 102 – Overall this evidence suggests

Line 103 – contributes should be contribute

Line 104 – their migration; NK cell

Line 105 – differentiation of pNK cells into dNK cells; mother/fetus should be maternal/fetal

Line 106 - contributes should be contribute

Line 107 – dNK cell

Line 107 – (eNK) cells 

Line 109 – dNK cells

Line 113 – ‘one of the first evidences’ should be ‘early evidence’

Line 114 – a uterine; dNK cells

Line 115 – Other evidence indicates 

Line 116 – spiral artery remodelling

Line 117 – dNK cells

Line 120 – dNK cell; collaborative network

Line 121 – results in endothelium development in the endometrium

Line 122 – dNK cell; has been shown

Line 123 – macrophage section needs mention of differences between resident populations of macrophages seeded in early development versus infiltrating macrophages derived from blood monocytes 

Line 126 – the main factors involved in macrophage differentiation are M-CSF (macrophage colony stimulating factor) plus other mediators that vary depending on the type of macrophage being generated.

Line 130 – two main types of macrophage:

Line 135 – endometrium has a low

Line 136 – changes across the menstrual; After fertilisation

Line 141 – the g in MIP-2g should be in Greek symbol

Line 142 – associated with human

Line 144 – GRO = growth regulated oncogene

Line 152 – have reduced expression of CD86 which is required for T lymphocyte activation

Line 153 – spiral artery remodelling

Line 154 – after that; concentration decreases

Line 157 – macrophages seem to

Line 159 – FGF = fibroblast growth factor

Line 162 – sentence starting ‘Recently, a new’ needs to be re-written

Line 168 – tree remodelling

Line 169 – prevent decidual release of; associated with accumulation of apoptotic bodies

Line 171 – macrophages plat a role in cervix remodelling 

Line 171 – and in the postpartum

Line 172 – uterus of senescent cells; contribute to recovery and maintenance of 

Line 174 – in this paragraph the authors mention potential use of macrophage targeted approaches for clinical trials but the examples focus on the neonate rather than elaborate on how these might be targeting the endometrium, decidua and placenta which are the focus of this section.

Line 177 – evidence is needed; of this knowledge through 

Line 181 – the placenta

Line 182 – fetal capillary network

Line 185 – maternal macrophages at the fetal-maternal interface

Line 186 – distinct origins that vary depending on

Line 188 – until parturition

Line 192 – and decline by term

Line 193 – A2M should be a2M

Line 200 – the arginine microenvironment is critical

Line 201 – Placental tissue 

Line 202 – the latter is required for polyamine production for cell proliferation

Line 203 – Therefore, placental arginine levels are low enough

Line 204 – soluble HLA-G which is

Line 211 – mesenchymal

Line 212 – Recent work reveals

Line 213 – isolated from placentas of women with gestational

Line 216 – hyporesponsive

Line 221 – FGF2, and culture media from these cells

Line 223 – cells promote branching

Line 223 - -3, and develop a mesenchymal-epithelial

Line 225 – It also has been suggested

Line 229 – (DCs)

Line 230 – the least well

Line 231 – between the innate and 

Line 232 – Typically, DCs take up bacteria and antigens 

Line 233 – which induces a maturation program that includes loss of endocytosis activity and up-regulated expression of co-activation molecules.

Line 235 – nodes for presenting antigen-derived peptides to host T cells in the context of major histocompatibility complex and drive a T helper

Line 236 – decidual DCs (dDCs)  

Line 237 – which are phenotypically 

Line 237 – DCs. Decidual DCs are

Line 238 – express combinations of 

Line 239 – all dDCs

Line 240 - T cell activating markers

Line 241 – From the early stages of implantation, migratory DCs populate

Line 243 – The angiogenic role of dDCs is mainly evident during implantation when they are most abundant. Their frequency declines progressively after that and they remain at low abundance throughout pregnancy.

Lines 247 & 248 – DCs

Line 249 – the sentence beginning Della Bella needs to be re-written to better convey the results from that work

Line 253 – peripheral; DCs

Line 255 – dDCs

Line 256 – immature characteristics of DCs.

Line 257 – DCs 

Line 258 – in diminished T cell

Line 263 – dehydroepiandrosterone

Line 268 – their ability to

Line 269 – constitutive immature DC 

Line 271 – Th2 has been used here but elsewhere type 2 has been used (e.g. line 260)

Line 272 – The abbreviation Tregs has not been defined earlier in the manuscript

Line 272 – NK cells

Line 277 – stroma and are mainly present 

Line 278 – DCs; within

Line 282 – are taken up by; DCs

Line 284 – DCs

Line 286 – dDCs; dNK cells

Line 287 – dDCs; innate immune defense. Decidual DCs are not effective at sensing bacteria and inducing T cell responses, although

Line 290 – need to highlight that lines 290 – 294 are about peripheral DCs

Line 295 – Elevated neutrophil numbers in blood is known as neutrophilia and there is a well-characterised increase in neutrophil counts 

Line 297 – In non-pregnant women around

Line 300 – understood by considering: a) neutrophils

Line 301 – b) neutrophils

Line 302 – combat tissue damage

Line 306 – c) neutrophils; DCs; and NK

Line 307 – highlight after full stop needs to be removed

Line 315 – interlaced network composed of

Line 318 – allows the catching of fungi,  

Line 319 – innate immune cells; NK cells

Line 322 – Microbicidal activity

Line 325 – too many spaces after glycoprotein

Line 328 – types, as mentioned previously, 

Line 331 – chemoattraction

Line 333 – tract tissue remodelling chemoattracts

Line 336 – arteries remodelling and

Line 338 – innate immune cells, the decidual 

Line 339 – phenotypic differences 

Line 339 – between decidual and peripheral neutrophils

Line 341 – confer a pro-angiogenic profile to neutrophils with enhanced 

Line 345  - Indeed

Line 347 – acts by

Line 350 – associated with excessive

Line 352 – in clinical practice

Line 353 – treatment of women

Line 358 – extra space after anaemia

Line 359 – responses

Line 360 – (TLRs)

Line 361 – domain. They belong

Line 361 – recognition receptor (PRR)

Line 366 – In mammals

Line 367 – functional TLRs

Line 367 – TLR nomenclature is usually TLR1 rather than TLR-1, etc

Line 369 – TLRs differ in 

Line 370 – signal via the canonical 

Line 372 – TLR3 used TRIF and TLR4 use TRIF and MyD88 so there needs to be some mention of this. This is mentioned later but should be highlighted earlier on as well.

Line 373 – explain TNFs better or use TNFa

Line 375 - such a model changes the concept of the innate immune response

Line 376 – and gives rise to specific signalling associated with a particular pathogen and the cell types 

Line 379 – for TLR-2

Line 381 – I think it is only TLR3 of these viral PAMP recognising TLRs that use TRIF (along with TLR4)

Line 389 – expression of TLRs

Line 390 – is summarised in table 1.

Line 392 – absent in term

Line 393 – TLR1/TLR2 recognises triacylated lipopeptides and TLR6/TLR2 recognises diacylated lipopeptides so this section needs to be re-written in light of this. In table 1 triacetylated and diacetylated need to be changed to triacylated and diacylated (TLR1, TLR2 and TLR10)

Line 396 – syncytiotrophoblasts

Line 399 – the syncytial layer

After table 1 line numbering from here

Line 10 – inefficient at controlling

Line 11 – actively

Line 12 – so can protect

Line 14 – As yet, there are no data

Line 17 – for TLR1 to TLR10 has

Line 20 – the need for

Line 22 – placenta replaces decidua 

Line 22 – pivotal role for

Line 24 – demonstrated

Line 25 – express all 10 TLR types

Line 26 – TLR4 expression on the apical side of the amniotic epithelium enables reaction to

Line 35 – associated with

Line 35 – not only is there a

Line 38 – we summarise the cytokine

Line 44 – what is meant by TLR4/TLR2 heterodimerisation patterns – this is a feature of TLR2 with TLR1 or TLR6 rather than TLR4

Line 44 – Another example of a 

Line 48 – magnified differently in response to a yeast

Table 2 heading – different TLR agonists 

Also Table 2 - Streptococcus

After Table 2 line numbering from here:

Line 1 – roles not related to 

Line 3 – canonical 

Line 6 – of TLR-dependent

Line 7 – induced pro-angiogenic

Line 8 – TLR5 is not considered a viral sensing TLR

Line 9 – Hence, deleterious

Line 11 – deserve

Line 13 – Local and systemic immune

Line 13 – establishing

Line 14 – dNK cells and macrophages

Line 18 – also phagocytose

Line 20 – T-lymphocyte

Line 21 – while the antigen presenting macrophages and dendritic cells

Author Response

Thank you for your comments concerning our manuscript ijms-513944 entitled "Innate immune cells and TLR-dependent responses at the maternal-fetal 
interface"
 by  Andrea Olmos-Ortiz, Pilar Flores-Espinosa, Ismael Mancilla-Herrera, Rodrigo Vega-Sanchez, Lorenza Díaz, Verónica Zaga-Clavellina.

We are very grateful for all reviewer’s comments and suggestions that have considerably improved the manuscript. We agree with your recommendation, therefore, we invited to Rodrigo Vega-Sánchez as a co-author, who worked reviewing the grammatical issues and the entire style to achieve a harmonic structure of the manuscript.

We have carefully reviewed each of your annotations, including that of improving the description of signaling pathways mediated by TLRs, where TRIF is activated by TLR3 and TLR4, while MyD88 is activated by the other TLRs including TLR4. In this revised version, a figure is included (Figure 2) to clarify this point.

Additionally, two new figures were included: Figure 1) Stages of recruitment and differentiation of decidual NK cells. Figure 3) Pro-angiogenic network mediated by innate immune cells at the maternal-fetal interface.

I hope these changes make our manuscript suitable for publication in the International Journal of Molecular Sciences.

I look forward to hearing from you at your earliest convenience.

Sincerely, 

Dr. Claudia Veronica Zaga Clavellina

Corresponding autor
Instituto Nacional de Perinatologia Isidro Espinosa de los Reyes
Montes Urales 800, Lomas Virreyes, D.F. 11000, México.

[email protected]

Reviewer 2 Report

In this Review, Olmos-Ortiz et al. introduce innate immune response at materno-fetal interface. The authors well summarized information about immune cells and TLRs in the decidua reported so far. Although this is a nice review, the authors should make an effort to make this review more informative.

In particular, it is strongly recommended that the authors make new histological graphics of the placenta/decidua to make all readers understand the location of immune cells and TLRs during pregnancy. In addition, the introduction of TLRs is too vague to comprehend the background of TLRs. The authors should rewrite the introduction section of TLRs. A concise and correct description of TLR would useful for readers.

the following points listed below should be taken into account to make this review refine.

Comments

1.       Line 360: Toll-like receptors (TLRs) are type I transmembrane proteins and can be subdivided to two types by those intracellular localization, cell surface TLRs & endosomal TLRs.

2.       There is no clear statement about TLR Ligands, the authors should explain “Table 1” and mention that TLR family sense three type of ligand, Nucleic Acid/Lipid/Protein, which correlate with intracellular localization.

3.       Line 364: Since there are no evidence that ROS itself work as DAMPs, “free radicals and reactive oxygen species (ROS)” should be removed.

4.       Lane366: Human has 10 TLRs (TLR1~TLR10) and one pseudogene TLR11, and mouse has 12 TLRs (TLR1~TLR13, except TLR10). So, “In the mammals,” is misleading and should be changed.

5.       Lane380/381: TLR family, except TLR3, uses the adaptor protein MyD88 for signal transduction. Additionally, TLR3 and TLR4 use the adaptor protein TRIF. “TLR-3,-7, … instead of the cellular membrane, and can also signal in a MyD88 independent manner,” is completely wrong statement

6.       Lane385/386: Genomic studies suggest that TLR10 is related to TLR1 and TLR6.

Author Response

Reviewer 2 Comments

In this Review, Olmos-Ortiz et al. introduce innate immune response at materno-fetal interface. The authors well summarized information about immune cells and TLRs in the decidua reported so far. Although this is a nice review, the authors should make an effort to make this review more informative.

In particular, it is strongly recommended that the authors make new histological graphics of the placenta/decidua to make all readers understand the location of immune cells and TLRs during pregnancy. In addition, the introduction of TLRs is too vague to comprehend the background of TLRs. The authors should rewrite the introduction section of TLRs. A concise and correct description of TLR would useful for readers.

The following points listed below should be taken into account to make this review refine.

Response to Reviewer 2 Comments

Thank you for your comments concerning our manuscript ijms-513944 entitled "Innate immune cells and TLR-dependent responses at the maternal-fetal 

interface"  by  Andrea Olmos-Ortiz, Pilar Flores-Espinosa, Ismael Mancilla-Herrera, Rodrigo Vega-Sanchez, Lorenza Díaz, Verónica Zaga-Clavellina.

We are very grateful for all reviewer’s comments and suggestions that have considerably improved the manuscript. We agree with your recommendation, therefore, we invited to Rodrigo Vega-Sánchez as a co-author, who worked reviewing the grammatical issues and the entire style to achieve a harmonic structure of the manuscript.

We have carefully reviewed each of your annotations, including that of improving the description of signaling pathways mediated by TLRs, where TRIF is activated by TLR3 and TLR4, while MyD88 is activated by the other TLRs including TLR4. In this revised version, a figure is included (Figure 2) to clarify this point.

In this revised version, we have included 3 new figures:

Figure 1) Stages of recruitment and differentiation of decidual NK cells.

Figure 2) TLRs signaling. MyD88-dependent and -independent pathways.

Figure 3) Pro-angiogenic network mediated by innate immune cells at the maternal-fetal interface.

We put particular effort for the improving of introductory section of TLRs and eliminate vague and ambiguous phrases.

Comments

1.      Line 360: Toll-like receptors (TLRs) are type I transmembrane proteins and can be subdivided to two types by those intracellular localization, cell surface TLRs & endosomal TLRs.

We agree with this comment, therefore, this description has now been included in lines 375-380.

2.      There is no clear statement about TLR Ligands, the authors should explain “Table 1” and mention that TLR family sense three type of ligand, Nucleic Acid/Lipid/Protein, which correlate with intracellular localization.

We included this suggestion in lines 378-380 and 387.

3.      Line 364: Since there are no evidence that ROS itself work as DAMPs, “free radicals and reactive oxygen species (ROS)” should be removed.

This suggestion has now been attended in the revised version.

4.      Lane366: Human has 10 TLRs (TLR1~TLR10) and one pseudogene TLR11, and mouse has 12 TLRs (TLR1~TLR13, except TLR10). So, “In the mammals,” is misleading and should be changed.

We agree with this comment and we have corrected this sentence in lines 376 - 378.

5.      Lane380/381: TLR family, except TLR3, uses the adaptor protein MyD88 for signal transduction. Additionally, TLR3 and TLR4 use the adaptor protein TRIF. “TLR-3,-7, … instead of the cellular membrane, and can also signal in a MyD88 independent manner,” is completely wrong statement

The reviewer is completely right. We have now changed the description of TLR signaling to an accurate version in lines 392-409. In this revised version, a new figure 2), TLRs signaling. MyD88-dependent and -independent pathways, is included to clarify this point.

6.       Lane385/386: Genomic studies suggest that TLR10 is related to TLR1 and TLR6.

This suggestion has now been referenced and included in lines 410-412.

I hope these changes make our manuscript suitable for publication in International Journal of Molecular Sciences.

I look forward to hearing from you at your earliest convenience.

Sincerely, 

Dr. Claudia Veronica Zaga Clavellina

Corresponding autor
Instituto Nacional de Perinatologia Isidro Espinosa de los Reyes
Montes Urales 800, Lomas Virreyes, D.F. 11000, México.

[email protected]

Round 2

Reviewer 2 Report

The authors responded to most of my concerns and those responses are reflected on revised version. In addition, I finally hope that the authors can respond to my comments below:

1, TLR4 cannot sense LPS/Lipid A without MD-2 which associate with TLR4 and directly bind Lipid A. (Kim et. al. Cell 2007, Ohto et. al. Science 2007)

2, TLR4 requires TIRAP and TRAM for MyD88 and TRIF signaling, respectively.

3, TLR7 and TLR9 induce IFN-α/β Production via MyD88/IRF7 pathway.

4, INF should be changed to IFN.

5, Most of TLRs make homodimers to sense ligands. TLR1 and TLR6 make heterodimers with TLR2 to sense those ligand.

Author Response

June 25, 2019

Dear Reviewer,

Thank you for your report concerning our manuscript ijms-513944 entitled "Innate immune cells and Toll-like receptor-dependent responses at the maternal-fetal 
interface"
 by  Andrea Olmos-Ortiz, Pilar Flores-Espinosa, Ismael Mancilla-Herrera, Rodrigo Vega-Sanchez, Lorenza Díaz, Verónica Zaga-Clavellina.

We deeply appreciate your comments for improving our manuscript. We requested a professional scientific English Edition in the order to improve our English language and style. In this version all changes were highlighted in yellow.

Comments and Suggestions for Authors

The authors responded to most of my concerns and those responses are reflected on revised version. In addition, I finally hope that the authors can respond to my comments below: 

 1, TLR4 cannot sense LPS/Lipid A without MD-2 which associate with TLR4 and directly bind Lipid A. (Kim et. al. Cell 2007, Ohto et. al. Science 2007)

We agree with the reviewer; therefore the changes suggested have been included in the  lines 407 – 410 of the manuscript. The recommended references have been also included.

2, TLR4 requires TIRAP and TRAM for MyD88 and TRIF signaling, respectively.

This suggestion has been taken into a count and incorporated to the manuscript in the line 410 and 420-422. And also it was included in the Figure 2.

3, TLR7 and TLR9 induce IFN-α/β Production via MyD88/IRF7 pathway.

We agree with the reviewer, therefore, it has now been mentioned in lines 426 and 427.

4, INF should be changed to IFN.

We revised all along the manuscript and the changes were made.

5, Most of TLRs make homodimers to sense ligands. TLR1 and TLR6 make heterodimers with TLR2 to sense those ligand.

Reviewer is right and now has been included in lines 399 – 402.

We are very grateful for all reviewer’s comments and suggestions that have considerably improved the manuscript.

We hope these changes make our manuscript suitable for publication in the International Journal of Molecular Sciences.

We look forward to hearing from you at your earliest convenience.

Sincerely, 

Dr. Claudia Veronica Zaga Clavellina

Corresponding author
Instituto Nacional de Perinatología Isidro Espinosa de los Reyes
Montes Urales 800, Lomas Virreyes, D.F. 11000, México.

[email protected]

Round 3

Reviewer 2 Report

The authors addressed all of my concerns. Finally, please take into account my minor comments below.

1, Descriptions in line 407~409, "however, an 407 exception is with TLR4 which needs Myeloid Differentiation Protein-2 (MD-2) for directly binding to the Lipid A core of lipopolysaccharide (LPS), and then the MD-2/LPS complex can be presented to TLR4 for its activation" are misleading.  TLR4 and MD-2 form preformed dimer without Lipid A. Lipid A binds to MD-2 complexed with TLR4 in CD14-dependent manner. In addition, MD-2 is not an abbreviation for “Myeloid Differentiation Protein-2”.